# Depth creates no more spurious local minima in linear networks

## Abstract

We show that for any convex differentiable loss, a deep linear network has no spurious local minima as long as it is true for the two layer case. This reduction greatly simplifies the study on the existence of spurious local minima in deep linear networks. When applied to the quadratic loss, our result immediately implies the powerful result by Kawaguchi (2016). Further, with the recent work by Zhou & Liang (2018), we can remove all the assumptions in (Kawaguchi, 2016). This property holds for more general "multi-tower" linear networks too. Our proof builds on the work in (Laurent & von Brecht, 2018) and develops a new perturbation argument to show that any spurious local minimum must have full rank, a structural property which can be useful more generally.

## 1 Introduction

One major mystery in deep learning is that deep neural networks do not seem to suffer from spurious local minima. Understanding this mystery has become one of the most important topics in the machine learning theory. This problem turns to be quite challenging. Even for the subcase of the deep linear network, much still remains unknown despite a long line of studies on this subject.

In this paper, we prove a useful structural property about the deep linear networks. We show that for any convex differentiable loss function, any spurious local minima in a deep linear network should already be present in a two layer linear network. Hence, depth does not create more spurious local minima in linear networks. This reduction greatly simplifies the study about the existence of spurious local minima in deep linear networks. When applied to the quadratic loss, it leads to the first unconditional proof that there is no local minima in deep linear networks.

Baldi & Hornik (1989) started the investigation on the existence of spurious local minima in linear networks. They showed that, under mild assumptions, for quadratic loss, two layer linear networks do not have spurious local minima. They also conjectured it is true for deep linear networks. This conjecture is only proved recently by Kawaguchi (2016). For the special case of linear residual networks, Hardt & Ma (2017) showed that there are no spurious local minima through a simpler argument.

While most existing work have been on quadratic loss functions, Laurent & von Brecht (2018) showed a surprisingly general result for any convex differentiable loss. In (Laurent & von Brecht, 2018), the authors consider the special linear networks which have no bottlenecks, i.e. when the narrowest layer is on the either end. They showed that for any convex differential loss function, there is no spurious local minima in such networks. In addition to its generality, the proof in (Laurent & von Brecht, 2018) is quite intuitive through a novel perturbation argument. However, their special cases excludes networks with bottleneck layers, commonly used in the practice and studied in the literature (Baldi & Hornik, 1989; Kawaguchi, 2016).

We build on the work in (Laurent & von Brecht, 2018) and further develop the technique to show that for general deep linear networks, whether there are spurious local minima is reduced to the two layer case.

**Theorem 1.** *Given any convex differentiable function $f : \mathbb{R}^{m \times n} \to \mathbb{R}$. For any $k \geq 2$, let $L_k(M_1, \ldots, M_k) = f(M_k \cdots M_1)$ where $M_i \in \mathbb{R}^{d_i \times d_{i-1}}$ for $1 \leq i \leq k$ with $d_k = m$ and $d_0 = n$. Let $d = \min_{0 \leq i \leq k} d_i$. Define $L_2(A, B) = f(AB)$ for $A \in \mathbb{R}^{m \times d}, B \in \mathbb{R}^{d \times n}$. Then $L_k$ has no spurious local minima iff $L_2$ has no spurious local minima.*

We emphasize that in the above theorem, $f$ depends on both the data and the loss function. Hence the reduction is instance specific and does not depend on the global property of a family of loss functions.

To prove Theorem 1, we show a non-degeneracy property of local minima, namely, any spurious local minimum must have rank same as the width of the bottleneck. This property can be useful for the two layer case too – it implies that each layer in a spurious local minimum cannot be rank-deficient, hence covering subcases which may otherwise require onerous analysis (Baldi & Hornik, 1989; Zhou & Liang, 2018).

**Theorem 2.** *With the same notation as in Theorem 1, if $A, B$ is a spurious local minimum of $L_2$, then both $A, B$ must have full rank.*

As an application of Theorem 1, for quadratic loss, Theorem 1, together with (Baldi & Hornik, 1989), immediately implies the main result in (Kawaguchi, 2016). We can further remove all the assumptions needed in (Baldi & Hornik, 1989; Kawaguchi, 2016), using the recent result by Zhou & Liang (2018) (Theorem 2(1)), hence providing the first unconditional proof of the non-existence of spurious local minima in deep linear networks for quadratic loss functions. Below $\|\cdot\|$ denotes the Frobenius norm.

**Corollary 1.** *For any $X \in \mathbb{R}^{d_0 \times n}, Y \in \mathbb{R}^{d_k \times n}$, let $L(M_1, \ldots, M_k) = \|M_k \cdots M_1 X - Y\|^2$. Then $L$ has no spurious local minima.*

Theorem 1 can be further generalized to "multi-tower" linear networks. Define a multi-tower linear network as the sum of multiple deep linear networks (towers), i.e. $M_{1k_1} \cdots M_{11} + \ldots + M_{sk_s} \cdots M_{s1})$, where $M_{ij} \in \mathbb{R}^{d_{i,j} \times d_{i,j-1}}$ with $d_{i,k_i} = m$ and $d_{i,0} = n$. For $1 \leq i \leq s$, let $b_i = \min_j d_{i,j}$ denote the bottleneck size of each tower $i$. Write $b = \sum_i b_i$.

**Corollary 2.** *For any differentiable convex loss $f$, a multi-tower linear network has no spurious local minima iff the linear network $AB$, where $A \in \mathbb{R}^{m \times b}, B \in \mathbb{R}^{b \times n}$, has no spurious local minima. Moreover, if $b \geq m, n$, there is no spurious local minima in this network.*

There are two main technical ideas in our proof. First, we show a reduction of any local minimum of the multi-layer network to a critical point of a two-layer network. Secondly, we generalize the local perturbation argument in (Laurent & von Brecht, 2018) to weaker critical conditions. Besides the conceptual importance of our results, these technical contributions might be useful for understanding deep networks too.

## 1.1 RELATED WORK

In Baldi & Hornik (1989), it is shown that, under mild assumptions on data, two layer linear network with quadratic loss has no spurious local minima. This is probably the first positive result on this long line of investigation. Kawaguchi (2016) showed that it holds for deep linear network too. The tour de force proof in (Kawaguchi, 2016) works by examining the Hessian using powerful tools from the matrix theory. There have been much subsequent work to simplify and generalize the result. For example, Lu & Kawaguchi (2017) came up with a different argument. Yun et al. (2018; 2019) showed simpler arguments for special cases and considered more general non-linear networks. Hardt & Ma (2017) showed that under certain assumptions, there might not even be stationary point in the deep linear residual network. Venturi et al. (2018) defined a notion of spurious valleys and showed that for quadratic losses, there is no spurious valley in deep linear networks. Venturi et al. (2018) was able to remove all the assumption in (Baldi & Hornik, 1989) under this weaker notion. The mild assumption in Baldi & Hornik (1989), which was also needed in Kawaguchi's proof, was removed by Zhou & Liang (2018), which leads to our Corollary 1.

Laurent & von Brecht (2018) considers the special case of linear networks with the narrowest layer on the either end. It uses a novel perturbation argument to show that for any convex differentiable loss, there is no spurious local minimum in such network. However, the special case considered in (Laurent & von Brecht, 2018) excludes networks through low rank approximation such as auto-encoders. But it is really the intuitive yet powerful result in (Laurent & von Brecht, 2018) that motivated this work.

There have been recent studies on the gradient descent convergence on the deep linear networks (Arora et al., 2018; Bartlett et al., 2018; Arora et al., 2019). It has been shown by Arora

et al. (2018) that, under certain conditions, increasing the depth of linear networks can speed up the convergence, which is another positive property of deep linear networks.

There have been much work (Soltanolkotabi et al., 2018; Li & Liang, 2018; Allen-Zhu et al., 2018a;b; Du et al., 2018a; Zou et al., 2018; Du et al., 2018b) recently on studying the optimization landscape and convergence of non-linear networks. They focus mostly on shallow networks with over-parameterized wide layers.

## 2 PRELIMINARIES

We define notations used through the paper. We state some simple facts and the main theorem from Laurent & von Brecht (2018) which we need in our proof.

**Deep linear networks.** Denote by $\mathbb{R}^{m \times n}$ all the matrices with $m$ rows and $n$ columns. For $1 \leq i \leq k$, let $M_i \in \mathbb{R}^{d_i \times d_{i-1}}$. For $i \geq j$, denote by $M_i \cdots M_j$ the matrix product of $M_i \cdot M_{i-1} \ldots M_j$. A (deep) linear network with parameters $M_1, \cdots, M_k$ is defined as $\Phi(x) = M_k \cdots M_1 x$. We call $k$ the depth of the network and $d_0, d_k$ the input and the output dimensions, respectively. Define $d = \min_{0 \leq i \leq k} d_i$ be the narrowest width. We say a network has a bottleneck if both $d_0 > d$ and $d_k > d$. A multi-tower linear network is defined as the sum of multiple linear networks (towers) with the same input and output dimensions.

**Empirical loss.** Given training data $D$ which consist of examples of pairs of $x, y$ where the input feature vector $x \in \mathbb{R}^{d_0}$, and the label $y$ in some arbitrary set, we wish to minimize the total loss:

$$L(M_1, \cdots, M_k; D) = \sum_{(x,y) \in D} f_y(\Phi(x)) = \sum_{(x,y) \in D} f_y(M_k \cdots M_1 x).$$

Define $f(A) = \sum_{(x,y) \in D} f_y(Ax)$. Then $L(M_1, \cdots, M_k; D) = f(M_k \cdots M_1)$. If $f_y$'s are all convex differentiable functions[1], then clearly $f$ is convex differentiable too. Below we omit $D$ and consider the loss function $L : \mathbb{R}^{d_1 \times d_0} \times \ldots \times \mathbb{R}^{d_k \times d_{k-1}} \to \mathbb{R}$ where $L(M_1, \cdots, M_k) = f(M_k \cdots M_1)$ for some $f : \mathbb{R}^{d_k \times d_0} \to \mathbb{R}$.

**Derivative.** Denote by $\frac{\partial L}{\partial M}$ the matrix form of the partial derivative of $L$ with respect to $M$. If $L$ has only one variable $M$, we write $L' = \frac{\partial L}{\partial M}$. For simplicity, we sometimes abuse the notation by using the same symbol for the variable and the value and omit the value. If $L(X, Y) = f(XY)$, by the chain rule, $\frac{\partial L}{\partial X} = f'(XY)Y^T$ and $\frac{\partial L}{\partial Y} = X^T f'(XY)$.

**Local minimum.** For any loss function $L$, $M_1, \ldots, M_k$ is a local minimum of $L$ if there exists an open ball $B$, in Frobenius norm, centered at $M_1, \ldots, M_k$ such that $L(M_1, \ldots, M_k) \leq L(M_1', \cdots, M_k')$ for any $(M_1', \cdots, M_k') \in B$. A local minimum is called *spurious* if it is not a global minimum. If $L$ is differentiable, then any local minimum is a critical point of $L$. In particular, if $L(M_1, \cdots, M_k) = f(M_k \cdots M_1)$, then $M_1, \cdots, M_k$ satisfy that $\frac{\partial L}{\partial M_i} = (M_k \cdots M_{i+1})^T f'(M_k \cdots M_1)(M_{i-1} \cdots M_1)^T = 0$

We need the following theorem from Laurent & von Brecht (2018):

**Theorem 3.** *Let $L_k(M_1, \ldots, M_k) = f(M_k \cdots M_1)$ where $f : \mathbb{R}^{d_k \times d_0} \to \mathbb{R}$ is a convex differentiable function. If there is no bottleneck, i.e. $d_0$ or $d_k = \min_{0 \leq i \leq k} d_i$, then any local minimum of $L_k$ is a global minimum of $f$.*

## 3 PROOFS

With the above preparation, we will now prove Theorem 1. If the network we consider has no bottleneck, i.e. $d = d_k$ or $d = d_0$, then Theorem 3 immediately implies that all the local minima

---

[1]In practice, $f_y$'s are typically convex. They are usually differentiable, and if not, can be smoothly approximated. For example the hinge loss can be approximated by the modified Huber loss (Zhang, 2004).

for $L_k$ are global minima of $f$ so the statement is vacuously true. Below we consider the case when $d = d_j$ for some $0 < j < k$. Let $A = M_k \cdots M_{j+1}$ and $B = M_j \cdots M_1$. The following is the main technical claim of the paper.

**Lemma 1.** *If $M_1, \ldots, M_k$ is a local minimum, then either $f'(AB) = 0$ or $A, B$ both have rank $d$.*

We first show that the above lemma implies Theorem 1.

*Proof.* **(Theorem 1)** If $L_2$ has spurious local minima, then $L_k$ has too. Since $d_j = \min_{0 \leq i \leq k} d_i$, for any $A \in \mathbb{R}^{d_k \times d_j}$ and $B \in \mathbb{R}^{d_j \times d_0}$, we can easily construct matrices $M_i \in \mathbb{R}^{d_i \times d_{i-1}}$ for $1 \leq i \leq k$ such that $M_k \cdots M_{j+1} = A$, and $M_j \cdots M_1 = B$. If $A, B$ is a spurious local minimum of $L_2$, then clearly $M_1, \cdots, M_k$ is a spurious local minimum of $L_k$.

The other direction is implied by Lemma 1. This implication has been used before multiple times by Kawaguchi (2016); Lu & Kawaguchi (2017); Yun et al. (2018). Here we include the easy proof for completeness. Suppose that $M_1, \cdots, M_k$ is a local minimum of $L_k$. Then by Lemma 1, either $f'(AB) = 0$ or $A, B$ both have rank $d$. If $f'(AB) = 0$, then $AB = M_k \cdots M_1$ is a global minimum of $f$ because $f$ is convex. Hence $M_1, \ldots, M_k$ is a global minimum of $L_k$ too.

In the other case, $A$ and $B$ both have full rank $d$. We show that any local perturbation to $A$ (resp. $B$) can be performed by local perturbation to $M_k$ (resp. $M_1$). If $A = M_k \cdots M_{j+1}$ has rank $d$, then $A_1 = M_{k-1} \cdots M_{j+1} \in \mathbb{R}^{d_{k-1} \times d}$ has rank $d$ too because $d \leq d_{k-1}$. Then for any $D \in \mathbb{R}^{d_k \times d}$, there exists $D_1 \in \mathbb{R}^{d_k \times d_{k-1}}$ such that $D_1 A_1 = D$. Hence $(M_k + D_1)A_1 = M_k A_1 + D_1 A_1 = A + D$. This implies any local perturbation to $A$ can be done through local perturbation to $M_1$. More precisely, there exists a constant $c > 0$, such that for any $D \in \mathbb{R}^{d_k \times d}$, there exists $D_1 \in R^{d_k \times d_{k-1}}$ with $\|D_1\| \leq c\|D\|$ and $D_1 A_1 = D$. Same is true for $B$. This implies that if $L_k(M_1, \ldots, M_k)$ is minimum in an open ball of radius $\delta$ centered at $M_1, \ldots, M_k$, then $L_2(A, B)$ is minimum in an open ball of radius $\delta/c$ centered at $A, B$. Hence if $M_1, \ldots, M_k$ is a local minimum of $L_k$, then $A, B$ is a local minimum of $L_2$. If $L_2$ has no spurious local minima, $A, B$, hence $M_1, \ldots, M_k$, is a global minimum. $\square$

In the above proof, we actually showed that if $M_1, \ldots, M_k$ is a spurious local minimum of $L_k$, then $A = M_k \cdots M_{j+1}, B = M_j \cdots M_1$ is a spurious local minimum of $L_2$. That is, every spurious local minima of $L_k$ can be directly mapped to a spurious local minimum of $L_2$, hence the title of the paper.

Lemma 1 directly implies Theorem 2.

*Proof.* **(Theorem 2)** Consider the case of $k = 2$ and $j = 1$. Then we have $A = M_2 \in \mathbb{R}^{d_2 \times d_1}$ and $B = M_1 \in \mathbb{R}^{d_1 \times d_0}$ with $d_1 < \min(d_0, d_2)$. If $A, B$ is a spurious local minimum of $L_2$, then $f'(AB) \neq 0$ because otherwise they would have been a global minimum of $f$. By Lemma 1, we have that both $A, B$ are of rank $d_1$, i.e. they both have full rank because $d_1 < d_0, d_2$. $\square$

To prove Lemma 1, we first observe that

**Lemma 2.** *If $M_1, \ldots, M_k$ is a local minimum of $L_k$, then $\frac{\partial L_2}{\partial A}(A, B) = 0, \frac{\partial L_2}{\partial B}(A, B) = 0$.*

*Proof.* Define $g_B(X) = L_2(X, B) = f(XB)$. Clearly $g_B$ is convex and differentiable too. Let $\widetilde{L}_B(M_{j+1}, \ldots, M_k) = g_B(M_k \cdots M_{j+1})$. If $M_1, \ldots, M_k$ is a local minimum of $L_k$, then $M_{j+1}, \ldots, M_k$ must be a local minimum of $\widetilde{L}_B$. In addition, $d_j = \min_{0 \leq i \leq k} d_i = \min_{j \leq i \leq k} d_i$, so there is no bottleneck in $M_{j+1}, \ldots, M_k$. We apply Theorem 3 to get that $A = M_k \cdots M_{j+1}$ is a global minimum of $g_B$, hence $\frac{\partial L_2}{\partial A}(A, B) = 0$. Similarly $\frac{\partial L_2}{\partial B}(A, B) = 0$. $\square$

Now we prove the key technical claim of Lemma 1.

*Proof.* **(Lemma 1)**

We just need to show that if $f'(AB) \neq 0$, then $A, B$ must be of rank $d$. Below we assume $f'(AB) \neq 0$. We will show that $A$ has rank $d$. For $B$, we can apply the same argument to $g(X) = f(X^T)$.

Let $r$ denote the rank of $A$. We will derive contradiction by assuming $r < d$. We first use an argument by Laurent & von Brecht (2018) to construct a family of local minima. Since $A = M_k \cdots M_{j+1}$ is of rank $r < d$, for any $2 \le i \le j + 1$, $M_k \cdots M_i$ has rank at most $r$. Since $r < d \le d_{i-1}$, there exists nonzero $w_{i-1} \in \mathbb{R}^{d_{i-1}}$ such that $M_k \cdots M_i w_{i-1} = 0$. Then for any $v_{i-1} \in \mathbb{R}^{d_{i-2}}$, we have

$$M_k \cdots M_i (M_{i-1} + w_{i-1} v_{i-1}^T) = M_k \cdots M_{i-1} \,.$$

Now for any $v_1, v_2, \cdots, v_j$ where $v_i \in \mathbb{R}^{d_{i-1}}$, we claim that

$$M_k \cdots M_{j+1} (M_j + w_j v_j^T) \cdots (M_1 + w_1 v_1^T) = M_k \cdots M_1 \,. \tag{1}$$

This can be shown inductively for $i = j, \cdots, 1$.

$$M_k \cdots M_{j+1} (M_j + w_j v_j^T) \cdots (M_i + w_i v_i^T) = M_k \cdots M_i$$

Since $M = (M_1, \ldots, M_k)$ is a local minimum, it is the minimum in an open neighborhood of $M$. If we set $\|v_i\|$'s small enough so that $\widetilde{M} = (M_1 + w_1 v_1^T, \cdots, M_j + w_j v_j^T, M_{j+1}, \cdots, M_k)$ is in a smaller neighborhood, then $\widetilde{M}$ is a local minimum too since $L_k(\widetilde{M}) = L_k(M)$. See Claim 1 in (Laurent & von Brecht, 2018) for a rigorous proof.

Let $\widetilde{B} = \widetilde{M}_j \cdots \widetilde{M}_1$. Then by Lemma 2, $\frac{\partial L_2}{\partial A}(A, \widetilde{B}) = 0$, i.e $\frac{\partial f(AB)}{\partial A}(A, \widetilde{B}) = f'(A\widetilde{B})\widetilde{B}^T = 0$. Since $A\widetilde{B} = AB$, we have that for any $\widetilde{M}_1, \ldots, \widetilde{M}_j$ constructed above,

$$f'(AB)\widetilde{B}^T = 0 \,. \tag{2}$$

For any matrix $M \in \mathbb{R}^{m_1 \times m_2}$, denote by $M^\ell \in \mathbb{R}^{m_2}$ the $\ell$-th row vector of $M$, and by $R(M)$ all the row vectors of $M$. Consider the linear subspace

$$V = \{ v \in \mathbb{R}^{d_0} \,|\, f'(AB)v = 0 \} \,.$$

Then (2) implies that $R(\widetilde{B}) \subseteq V$. We now show that we can choose $v_i$'s for $1 \le i \le j$, such that $\widetilde{B} = \widetilde{M}_j \cdots \widetilde{M}_1$ contains a row vector which is not in $V$ to reach a contradiction.

Let $i^* = \min\{i \,|\, R(M_i \cdots M_1) \subseteq V\}$. If $i^* = 1$, we choose a sufficiently small non-zero vector $v_1 \notin V$. This can be done by our assumption that $f'(AB) \ne 0$. Set $\widetilde{M}_1 = M_1 + w_1 v_1^T$. Since $w_1 \ne 0$, there exists $\ell$ such that $w_{1\ell} \ne 0$. Then $\widetilde{M}_1^\ell = M_1^\ell + w_{1\ell} v_1$. By $M_1^\ell \in V, v_1 \notin V, w_{1\ell} \ne 0$, we have $\widetilde{M}_1^\ell \notin V$ since $V$ is a linear subspace. Now assuming $i^* > 1$. Suppose that we have constructed $\widetilde{M}_i, \ldots, \widetilde{M}_1$, for some $i \ge i^* - 1 \ge 1$, such that $R(\widetilde{M}_i \cdots \widetilde{M}_1) \nsubseteq V$. We show the construction for $i + 1$. If $R(M_{i+1} \widetilde{M}_i \cdots \widetilde{M}_1) \nsubseteq V$, then we can simply set $\widetilde{M}_{i+1} = M_{i+1}$. Assume below that $R(M_{i+1} \widetilde{M}_i \cdots \widetilde{M}_1) \subseteq V$. By inductive hypothesis $R(\widetilde{M}_i \cdots \widetilde{M}_1) \nsubseteq V$, thus there exists say the $\ell$-th row vector $v$ of $\widetilde{M}_i \cdots \widetilde{M}_1$ not in $V$. Set $v_{i+1}$ as the $\ell$-th basis vector in $R^{d_i}$ so that $v_{i+1}^T \widetilde{M}_i \cdots \widetilde{M}_1 = v^T$. Now let $\widetilde{M}_{i+1} = M_{i+1} + w_{i+1} v_{i+1}^T$. Then

$$\widetilde{M}_{i+1} \widetilde{M}_i \cdots \widetilde{M}_1 = M_{i+1} \widetilde{M}_i \cdots \widetilde{M}_1 + w_{i+1} v_{i+1}^T \widetilde{M}_i \cdots \widetilde{M}_1$$
$$= M_{i+1} \widetilde{M}_i \cdots \widetilde{M}_1 + w_{i+1} v^T \,.$$

Since $R(M_{i+1} \widetilde{M}_i \cdots \widetilde{M}_1) \subseteq V$ but $v \notin V$ and $w_{i+1} \ne 0$, by the same argument as for $i^* = 1$, there must exist a row vector in $\widetilde{M}_{i+1} \cdots \widetilde{M}_1$ which is not in $V$. We have inductively constructed $\widetilde{B} = \widetilde{M}_j \cdots \widetilde{M}_1$ such that $R(\widetilde{B}) \nsubseteq V$, contradicting to (2). Hence $A$ must have rank $d$. This concludes the proof. $\qquad\square$

Corollary 1 immediately follows from Theorem 1 and Theorem 2(1) in (Zhou & Liang, 2018). In the following, we prove Corollary 2.

*Proof.* (**Corollary 2**) If some tower has no bottleneck, we can fix all the parameters but this tower, we can then apply Theorem 1 to show that any local minimum is a global minimum. Hence the statement holds. Below we assume that each tower has a bottleneck with width $b_i$.

Suppose that we have a spurious local minimum $\mathcal{M} = (M_{11}, \cdots, M_{1k_1}, \cdots, M_{s1}, \cdots, M_{sk_s})$. Similar to the proof of Theorem 1, we can break each tower $i$ as $A_i \in \mathbb{R}^{m \times b_i}, B_i \in \mathbb{R}^{b_i \times n}$ at the bottleneck layer. Write $M = \sum A_i B_i$. Similarly we can show that either $f'(M) = 0$ or all the $A_i, B_i$'s have full rank. Since $\mathcal{M}$ is a spurious local minimum, it must be the second case, i.e. all the $A_i, B_i$ have full rank. Now let $A = (A_1, \cdots, A_s)$ and $B = \begin{pmatrix} B_1 \\ \vdots \\ B_s \end{pmatrix}$. Then $A \in \mathbb{R}^{m \times b}, B \in \mathbb{R}^{b \times n}$ where $b = b_1 + \cdots + b_s$. By the same argument in the proof of Theorem 1, any perturbation of $A, B$ can be done through perturbation of $\mathcal{M}$, hence $A, B$ is a spurious local minimum for the single tower two layer network $AB$. If $b \geq m, n$, then Theorem 3 implies that any local minimum is a global minimum of $f$. $\square$

## 4 CONCLUSION

We have shown a non-degeneracy property of local minima in deep linear networks for general convex differentiable loss. This property allows us to reduce the existence of spurious local minima in a deep (with depth $\geq 3$) linear network to the two layer linear network, and, for two layer networks, to simplify analysis by removing the rank-deficient case. We show the application to quadratic loss functions and the generalization to multi-tower deep linear networks. Our proof uses a novel perturbation argument and does not require any heavy mathematical machinery.

It would be interesting to study when there is no spurious local minima beyond the quadratic loss. By our result, we only need to consider the two layer case. Another interesting question is whether similar phenomenon exists for non-linear networks.

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
