# OpenReview forum: "Depth creates no more spurious local minima in linear networks"
_ICLR.cc/2020/Conference — Reject_

### Official Review · AnonReviewer1 · 2019-10-16
**Official Blind Review #1**

**Rating:** 3

**Review:**

Summary:

The paper shows a deep linear network has no spurious local minima as long as it is true for the two layer case for any convex differentiable loss.

Comments:

1) I understand that there exists some work on deep linear network recently. However, they seem to be only for theoretical purpose. Most of the current practical problems do not consider this kind of network for training. If it has high impact in practice, then people are starting to use it. Could you please provide more reasons why we need to care about this impractical network?

2) It is still unclear about the contributions of the paper. Why “deep linear network has no spurious local minima as long as it is true for the two layer case” is important? And what we can take any advantage from here? What if there exist some spurious local minima for the two layer case (which is widely true)?

3) The paper looks like a technical report and seems not to be ready.

The results are quite incremental from the existing ones. The contributions of this work to the deep learning community are still ambiguous.


**Experience Assessment:**

I have read many papers in this area.

**Review Assessment: Checking Correctness Of Derivations And Theory:**

I assessed the sensibility of the derivations and theory.

**Review Assessment: Checking Correctness Of Experiments:**

N/A

**Review Assessment: Thoroughness In Paper Reading:**

N/A

---

> ### Author Response · Authors · 2019-11-07
> **Author response**
>
> Thank you very much for your comments, which is very helpful for clarifying our contribution and improving the presentation of the paper. Please see the inline responses.
>
> > Comments:
> >
> > 1) I understand that there exists some work on deep linear network recently. However, they seem to be only for theoretical purpose. Most of the current practical problems do not consider this kind of network for training. If it has high impact in practice, then people are starting to use it. Could you please provide more reasons why we need to care about this impractical network?
>
> It is true that deep linear networks are not used much, if any, in practice (though two layer linear networks, e.g. matrix factorization, is broadly used for recommender systems). The main reason to study this is to gain insight and to invent tools to understand the practical case. This is quite common practice (for example, recent studies on wide shallow networks) when we are unable to solve the eventual question but we would still like to make progress.
>
> > 2) It is still unclear about the contributions of the paper. Why “deep linear network has no spurious local minima as long as it is true for the two layer case” is important? And what we can take any advantage from here?
>
> Both nonlinearity and depth can increase the complexity of the optimization landscape. It would be super interesting to show that depth does not hurt. Our paper showed that this is the case for deep linear networks. This is an interesting conceptual contribution. The proof requires a few novel arguments too.
>
> > What if there exist some spurious local minima for the two layer case (which is widely true)?
>
> We know they do not exist for quadratic loss and  for any differentiable convex loss when there is no bottleneck. Actually we were unable to construct an example (although we suspect they do exist).
>
> > 3) The paper looks like a technical report and seems not to be ready.
>
> The paper was intended as a clean proof of a clean statement. Your (and others) comments have been very useful for clarifying the contribution of the paper and improving the presentation. We would appreciate any further advices on what to include in the paper.
>
> > The results are quite incremental from the existing ones. The contributions of this work to the deep learning community are still ambiguous.
>
> Besides the contribution stated above, with our paper, we now know that there is no spurious local minima in deep linear network for quadratic losses (this was only known conditionally before our paper). In addition, our proof is quite accessible which we hope to help to enable further progress on related topics.

---

### Official Review · AnonReviewer3 · 2019-10-22
**Official Blind Review #3**

**Rating:** 6

**Review:**

The paper shows an interesting result: deep linear NN has introduced no more spurious local minima than two layer NN and provides an intuitive and short proof for the results, which improve and generalize the previous results under milder assumptions. Overall, the paper is well written and clear in comparison and explanation.

The weakness is that the main theoretical contribution seems to be merely Lemma 1, and all other theorems are a direct corollary. Also, it would be of great interest to see concrete results on non-linear neural networks, since that is exactly what is used in common practice.


**Experience Assessment:**

I have read many papers in this area.

**Review Assessment: Checking Correctness Of Derivations And Theory:**

I assessed the sensibility of the derivations and theory.

**Review Assessment: Checking Correctness Of Experiments:**

I assessed the sensibility of the experiments.

**Review Assessment: Thoroughness In Paper Reading:**

I read the paper at least twice and used my best judgement in assessing the paper.

---

> ### Author Response · Authors · 2019-11-07
> **Author response**
>
> Thank you very much for your comments. We agree with you that the non-linear neural networks is the ultimately interesting question. We hope the work here can provide some new angle (e.g. the reduction from deep to two layer networks) towards that question.

---

### Official Review · AnonReviewer2 · 2019-10-27
**Official Blind Review #2**

**Rating:** 3

**Review:**

The motivation of this paper is training deep neural network seems to not suffer from local minima, and it tries to explain this phenomenon by showing that all local minima of deep neural network is global minima. The paper shows that for any convex differentiable loss function, a deep linear neural network has no so called spurious local minima, which to be specific, are local minima that are not global minima, as long as it is true for two-layer Neural Network. The motivation is that combining with existing result that no spurious local minima exists for quadratic loss in two-layer Neural Network, this relation connecting between two-layer and deeper linear neural network immediately implies an existing result that all local minima are global minima, removing all assumptions. The result also holds for general “multi-tower” linear networks.

Overall, this paper could be an improvement of existing results. It is well written and the proof step is clear in general. However, there’re some weakness need clarifications on the results, especially on the novelty. Given reasonable clarifications in response, I would be willing to change my score.

For novelty, it is unclear if the results from Lemma 1 to Theorem 1 and 2 are both being stated as novel results. The first part of proof of Theorem 1 is obvious and straightforward, and the other direction has been used before for multiple times as claimed in the paper, what is your novelty exactly here? For the key technical claim of Lemma 1, it looks like this perturbation technique already exists in (Laurent & Brecht, 2018), why do you claim it as a novel argument?

Besides novelty, there are also some other unclear pieces in this paper needs clarification:
1)	Is the main result which is “no spurious local minima for deep neural network” holds for any differentiable convex loss other than quadratic loss? How will Theorem 1 help us understand the mystery of neural network?
2)	How does the result help us understand non-linear deep neural network, which is commonly use in practice?
3)	The paper should give some explanations about why the results help training neural networks.


**Experience Assessment:**

I have read many papers in this area.

**Review Assessment: Checking Correctness Of Derivations And Theory:**

I carefully checked the derivations and theory.

**Review Assessment: Checking Correctness Of Experiments:**

N/A

**Review Assessment: Thoroughness In Paper Reading:**

I read the paper thoroughly.

---

> ### Author Response · Authors · 2019-11-07
> **Author response**
>
> Thank you very much for your detailed review and the comments/questions, which are very helpful for clarifying our contribution and improving the presentation of the paper.
>
> Please see the inline responses to your specific questions.
>
> > For novelty, it is unclear if the results from Lemma 1 to Theorem 1 and 2 are both being stated as novel results. The first part of proof of Theorem 1 is obvious and straightforward, and the other direction has been used before for multiple times as claimed in the paper, what is your novelty exactly here?
>
> The main technical contribution is Lemma 1. The other claims (Theorem 1, 2, Cor 1) follow more or less directly from it, but they are interesting conceptually. We did state explicitly in the proof of Theorem 1 that the implication of Lemma 1 to Theorem 1 (a rather easy argument) is included for the completeness.  The fact that the property of deep linear networks can be determined by the two layer network is certainly novel and interesting too (in our opinion). Besides the conceptual novelty, there is also technical novelty, as stated below.
>
> > For the key technical claim of Lemma 1, it looks like this perturbation technique already exists in (Laurent & Brecht, 2018), why do you claim it as a novel argument?
>
> The proof is inspired by Laurent & Brecht, 2018, as explicitly stated in the paper. Especially, it follows the same argument to construct a family of local minima (up to line 14 on page 5.) But then the proof branches from there. In Laurent&Brecht, the critical condition used is that the null space is the entire space. So it only requires one line (formula (21) in that paper) to carry the induction. But here, because of the existence of bottleneck, the critical condition is that the local minima must lie on some subspace. It requires to generalize the argument to deal with this case. The bulk of the proof of Lemma 1 (from line 15 on) is about carrying the induction through with this weaker constraints. But we should contrast this better in the paper.
>
> > Besides novelty, there are also some other unclear pieces in this paper needs clarification:
> > 1)	Is the main result which is “no spurious local minima for deep neural network” holds for any differentiable convex loss other than quadratic loss?
>
> The main result is as stated in Theorem 1, i.e.  for any differentiable convex loss, whether a deep linear network has spurious local minima reduces to the two layer case.
>
> > How will Theorem 1 help us understand the mystery of neural network?
>
> Conceptually, it shows that depth does not introduce extra local minima for deep linear networks. In general, the complexity of landscape can be caused by the non-linearity and/or depth. Here we show that depth does not make the landscape more complex for the linear networks. We think this is a progress towards understanding the mystery of neural networks. And if not,  any understanding of deep neural network should include deep linear network as a special case. So it is a pre-requisite to some sense.
>
> > 2)	How does the result help us understand non-linear deep neural network, which is commonly use in practice?
>
> Good question. We can only speculate here --- perhaps similar phenomena exist for non-linear networks? It would greatly simplify our task if we can reduce the study to two or small number of layers. We have shown this is possible for deep linear networks, which is perhaps interesting, at least conceptually?
>
> > 3)	The paper should give some explanations about why the results help training neural networks.
>
> The paper is solely about understanding the landscape of deep linear networks, which is, in our opinion, an important question which needs to be answered even before studying the convergence.

---

### Public Comment · ~Micah_Goldblum1 · 2019-11-08
**An Interesting Connection**

Hi Authors,
Thank you for your interesting paper.  I wanted to bring to your attention that your insights into spurious local minima is related to our paper which shows, both theoretically and empirically, that highly suboptimal local minima do exist in the loss landscape of nonlinear neural networks.[1]  Please consider mentioning the relationship with our work in your next version.

[1] https://arxiv.org/abs/1910.00359

---

> ### Author Response · Authors · 2019-11-12
> **Thanks**
>
> Thanks for the reference!

---

### Decision · Program_Chairs · 2019-12-19

**Decision:**

Reject

**Comment:**

Paper shows that the question of linear deep networks having spurious local minima under benign conditions on the loss function can be reduced to the two layer case. This paper is motivated by and builds upon works that are proven for specific cases. Reviewers found the techniques used to prove the result not very novel in light of existing techniques. Novelty of technique is of particular importance to this area because these results have little practical value in linear networks on their own; the goal is to extend these techniques to the more interesting non-linear case.